# Secretome Analysis of *Arabidopsis*–*Trichoderma atroviride* Interaction Unveils New Roles for the Plant Glutamate:Glyoxylate Aminotransferase GGAT1 in Plant Growth Induced by the Fungus and Resistance against *Botrytis cinerea*

**DOI:** 10.3390/ijms22136804

**Published:** 2021-06-24

**Authors:** María del Carmen González-López, Saúl Jijón-Moreno, Mitzuko Dautt-Castro, Cesaré Ovando-Vázquez, Tamar Ziv, Benjamin A. Horwitz, Sergio Casas-Flores

**Affiliations:** 1Laboratorio de Genómica Funcional y Comparativa, División de Biología Molecular, IPICYT, Camino a la Presa San José No. 2055. Col. Lomas 4ª. Section, San Luis Potosí C.P. 78216, Mexico; maria.gonzalez@ipicyt.edu.mx (M.d.C.G.-L.); saul.jijon@ipicyt.edu.mx (S.J.-M.); mitzuko.dautt@ipicyt.edu.mx (M.D.-C.); cesare.ovando@ipicyt.edu.mx (C.O.-V.); 2Centro Nacional de Supercómputo, IPICYT, Camino a la Presa San José No. 2055. Col. Lomas 4ª. Section, San Luis Potosí C.P. 78216, Mexico; 3Smoler Protein Center, Faculty of Biology, Technion—Israel Institute of Technology, Haifa 32000, Israel; tamarz@technion.ac.il; 4Faculty of Biology, Technion—Israel Institute of Technology, Haifa 32000, Israel; horwitz@technion.ac.il

**Keywords:** *Arabidopsis*, *Trichoderma*, secretome, enzymes, glycosidases, GGAT1, plant growth, induced systemic resistance, hydrogen peroxide

## Abstract

The establishment of plant–fungus mutualistic interaction requires bidirectional molecular crosstalk. Therefore, the analysis of the interacting organisms secretomes would help to understand how such relationships are established. Here, a gel-free shotgun proteomics approach was used to identify the secreted proteins of the plant *Arabidopsis thaliana* and the mutualistic fungus *Trichoderma atroviride* during their interaction. A total of 126 proteins of *Arabidopsis* and 1027 of *T. atroviride* were identified. Among them, 118 and 780 were differentially modulated, respectively. Bioinformatic analysis unveiled that both organisms’ secretomes were enriched with enzymes. In *T. atroviride*, glycosidases, aspartic endopeptidases, and dehydrogenases increased in response to *Arabidopsis*. Additionally, amidases, protein-serine/threonine kinases, and hydro-lyases showed decreased levels. Furthermore, peroxidases, cysteine endopeptidases, and enzymes related to the catabolism of secondary metabolites increased in the plant secretome. In contrast, pathogenesis-related proteins and protease inhibitors decreased in response to the fungus. Notably, the glutamate:glyoxylate aminotransferase GGAT1 was secreted by *Arabidopsis* during its interaction with *T. atroviride*. Our study showed that GGAT1 is partially required for plant growth stimulation and on the induction of the plant systemic resistance by *T. atroviride*. Additionally, GGAT1 seems to participate in the negative regulation of the plant systemic resistance against *B. cinerea* through a mechanism involving H_2_O_2_ production.

## 1. Introduction

In their natural settings, plants interact with a plethora of microorganisms, including bacteria, oomycetes, and fungi, through dynamic interactions ranging from parasitism to mutualism [1]. Plants restrict the access of invading microorganisms through mechanical structural barriers such as cell walls, waxes, and trichomes [2]. Furthermore, plants can detect invaders responding with the accumulation of toxic chemical compounds including alkaloids, phenols, terpenoids, and glucosinolates [3]. Phytopathogens secrete an array of lytic enzymes into the extracellular space to degrade the cell wall polymers including cellulose, xylan, and pectin to surpass the plant cell wall [4,5].

Once pathogens overcome the plant’s physical and chemical barriers, they are recognized through inducible defense mechanisms, including the perception of their highly conserved pathogen- or microbe-associated molecular patterns (PAMPs or MAMPs) [6]. This perception is carried out by membrane-associated receptors, called pattern recognition receptors (PRRs), which lead to the activation of the first layer of defense, termed PAMP-triggered immunity (PTI) that wards off invading microorganisms [7,8]. Successful pathogens secrete a repertoire of effector molecules, some of which are delivered into the plant cell facilitating the infection [9]. Intracellular effectors interfere with immune signaling leading to the so-called effector-triggered susceptibility (ETS). Nevertheless, plants have developed strategies to to directly or indirectly detect effectors through intracellular receptors termed nucleotide-binding, leucine-rich repeat receptors (NLRs). Consequently, the plant cells activate a second layer of defense named effector-triggered immunity (ETI), which is generally associated with the hypersensitive response (HR) [6]. The stimulation of PTI leads to the activation of several physiological events, such as the accumulation of reactive oxygen species (ROS), callose deposition, biosynthesis of antimicrobial metabolites, and the production of a complex blend of phytohormones, including salicylic acid (SA), jasmonate (JA), and ethylene (ET), among others, which act as signaling molecules [2,10]. Upon local activation of PTI or ETI, some of two forms of systemic resistance are activated in the plant: systemic acquired resistance (SAR) and induced systemic resistance (ISR). SAR signaling is dependent on SA and is involved in defense against biotrophic and hemibiotrophic pathogens (i.e., *Pseudomonas syringae*). ISR requires JA and ET as signaling molecules, and it is involved in response against necrotrophic microorganisms (i.e., *Botrytis cinerea*), herbivores [10], and beneficial microbes [6].

During plant–microbe interaction, plants secrete an array of proteins to the extracellular space, which is commonly referred to as the apoplast, where they participate in cell wall structure maintenance and potentially in the establishment of a molecular dialog with the invaders, leading to different types of relationships [11]. The whole set of proteins secreted by the host plant or its microbial partner constitutes their so-called secretome, whose constituent proteins are secreted at a particular time by known or unknown mechanisms of transport [11]. Many of these proteins are conventionally transported through the endoplasmic reticulum (ER) to the Golgi and then to the plasma membrane, where they are finally secreted into the apoplast. Conventionally secreted proteins contain an N-terminal signal peptide (SP). Proteins that lack an SP reach the extracellular space using unconventional protein secretion (UPS) pathways [12]. Some proteins secreted by UPS do so by a non-vesicular means of transport, bypassing the Golgi. Thus, proteins are directly secreted and translocated across the plasma membrane [12]. Other means of transport by UPS depend on specific intracellular compartments including vacuoles and exocyst-positive organelle (EXPO) [13].

Mutualistic fungi provide multiple benefits to their host plants, such as stimulation of plant growth and protection against abiotic and biotic stresses [14,15]. Some examples of mutualistic fungi include the mycorrhizal fungi *Glomus intraradices* and *Laccaria bicolor* and the plant growth-promoting fungi *Trichoderma* spp. [14,15,16]. *Trichoderma* spp. are filamentous ascomycetes found as free-living microorganisms growing on decomposed organic matter. Members of this genus colonize the roots of a broad range of plants [15]. The mechanisms by which *Trichoderma* may influence plant growth include the synthesis of phytohormones [17], the solubilization of phosphate [18], and the production of secondary metabolites [19,20]. Furthermore, plants colonized by *Trichoderma* spp. induce the SAR and ISR simultaneously and transiently, mounting a physiological phenomenon called priming [21]. Priming is characterized by an enhanced activation of induced defense mechanisms at distal parts of the infection, resulting in a faster and more effective response upon a second infection by phytopathogens [15,22,23]. Furthermore, *Trichoderma* spp. induce the expression of genes related to defense, such as the plant defensin 1.2 gene (*PDF1.2*) and the pathogenesis-related 1a gene (*PR-1a*), which are used as markers for ISR-priming and SAR, respectively [19,22,23,24,25]. In addition, some proteins of *Trichoderma* spp. (i.e., small-secreted cysteine-rich proteins (sm1), cellulases (Thp1 and Thp2), xylanases (Xyn2/Eix)) and volatile organic compounds (i.e., 6-pentyl-2H-pyran-2-one, 2-heptanone, 3-octanol) act as elicitors of plant defense responses [15,19,21]. These findings, along with results derived from our research group (unpublished data), indicate that *Trichoderma* is initially recognized as a pathogen, later suppressing the plant immunity to establish a mutualistic association with its host plant [23].

Proteomic studies have provided of new insights into the molecular crosstalk that occurs between *Trichoderma* spp. and their host plant. For instance, *T. virens* secretes an array of proteins potentially involved in cell-wall degradation, scavenging of ROS and secondary metabolism during its interaction with maize plants [26,27]. In response to *T. virens*, maize plants secrete into the apoplast proteins related to defense, including proteinase inhibitors and PR proteins [26]. Interestingly, *Arabidopsis* secretes photosynthetic proteins, including the glutamate:glyoxylate aminotransferase 1 protein, GGAT1, in extracellular vesicles (EVs) [28]. However, the role of most of these proteins in the establishment of a mutualistic relationship between both organisms remains unclear.

Here, to unravel the possible role of *A. thaliana* (hereafter referred to as *Arabidopsis*) and *Trichoderma atroviride* secreted enzymes in the establishment of a mutualistic relationship, their secretomes were assessed at different times of interaction. *Arabidopsis* and *T. atroviride* were grown alone or in co-culture in a semi-hydroponic system, their secretomes were obtained, and their secreted proteins were identified by liquid chromatography-tandem mass spectrometry (LC-MS/MS). Subsequently, a quantitative proteomic analysis was performed to identify the differentially accumulated proteins of both organisms. Furthermore, several bioinformatics tools were used to predict the protein functions that potentially take part in the establishment of such mutualistic association. Finally, GGAT1, which was differentially accumulated, was chosen to assess its role in plant growth stimulation and in the triggering of systemic resistance by *T. atroviride* against the phytopathogenic fungus *B. cinerea*.

## 2. Results

### 2.1. Time-Course Analysis of the Arabidopsis–T. atroviride Interaction Secretome

To dissect the repertoire of proteins secreted by *Arabidopsis* and *T. atroviride* during their interaction in a semi-hydroponic system at 24, 48, and 96 h of co-culture (Appendix A), samples of two independent biological replicates were analyzed by LC-MS/MS. A total of 1153 proteins were identified. Mapping of the predicted peptides showed that 1027 proteins pertained to the fungus, whereas 126 belonged to the plant (Appendix A).

Furthermore, we analyzed the secretomes changes to identify proteins that potentially could be involved in the establishment of the interaction at early (24 and 48 h) and late (96 h) times. The *Arabidopsis* or *T. atroviride* proteins were filtered to include only those classified as differentially accumulated (log_2_ ≥ 2.0 or ≤−2.0) with a false discovery rate (FDR) <0.01, compared to *Arabidopsis* or *T. atroviride* controls growing alone. Furthermore, the differentially accumulated proteins must be present in at least one of the three-time points of co-culture in both biological replicates. Accordingly, 118 proteins of *Arabidopsis* were differentially accumulated, of which 78 increased whereas 40 decreased. On the *T. atroviride* side, 780 proteins were differentially accumulated, of which 477 increased, whereas 303 decreased (Appendix A). As shown in Figure 1, most *Arabidopsis* and *T. atroviride* proteins were differentially modulated at 48 and 96 h of co-culture, respectively.

To determinate the common and unique proteins of *Arabidopsis* and *T. atroviride* with increased or decreased abundance over time, they were grouped using Venn diagrams. Only 20 (25.6%) of the increased and nine (22.5%) of the decreased proteins from *Arabidopsis* overlapped at all times of interaction (Figure 2A). In the *T. atroviride* secretome, only 24 (5%) of the increased and 10 (3.3%) of the decreased proteins overlapped at all times of co-culture, while unique proteins at each time analyzed were more at specific time points (Figure 2B).

### 2.2. Arabidopsis and T. atroviride Proteins Are Conventionally and Unconventionally Secreted during their Symbiosis

The differentially modulated secreted proteins of *Arabidopsis* and *T. atroviride* were categorized based on their predicted secretion pathways. We found that 84 proteins (71%) of *Arabidopsis* and 314 (40%) of *T. atroviride* were predicted to be conventionally secreted. In addition, 30 proteins (25.5%) of the plant and 331 (43%) of the fungus were predicted as secreted by unconventional pathways. The remaining four proteins (3.5%) of *Arabidopsis* and 135 (17%) of *T. atroviride* were predicted as not secreted (Figure 3A, Figure 4A, and Appendix A).

### 2.3. Functional Annotation of Arabidopsis and T. atroviride Secreted Proteins

To better understand the diverse repertoire of *Arabidopsis* and *T. atroviride*-secreted proteins, a Gene Ontology (GO) analysis was performed, including cellular component, biological process, and molecular function terms (Appendix A). Prediction of cellular localization for all plant modulated proteins pointed mainly to the extracellular region, plasma membrane, and cytoplasm. Proteins derived from secretory vesicles and the apoplast were also detected in the *Arabidopsis* secretome (Figure 3B). The most representative cellular components for *T. atroviride*-secreted proteins were the extracellular region, plasma membrane and cytoplasm. A small subset of *T. atroviride* proteins was predicted to localize in other cellular components, such as the endoplasmic reticulum, multivesicular body, and nucleus (Figure 4B). According to this analysis, *Arabidopsis* and *T. atroviride* secretomes were enriched mainly with extracellular proteins; however, a small subset of proteins probably derived from the plasma membrane and intracellular origin (Figure 3B and Figure 4B).

In *Arabidopsis*, the differentially modulated proteins were mainly classified in biological processes with potential functions related to defense response, oxidation–reduction, proteolysis, and response to oxidative stress processes (Figure 3C). For the molecular function term, proteins were predicted to have chitinase, peroxidase, oxidoreductase, and hydrolase activities, among others (Figure 3D). No biological process was predicted for 13 proteins, nor was a molecular function predicted for 16 proteins of *Arabidopsis* (Appendix A). Regarding the biological process term, *T. atroviride* proteins were mainly classified in oxidation–reduction, proteolysis, carbohydrate metabolism, chitin catabolism, and cellulose catabolism (Figure 4C). For the molecular function term, the *T. atroviride* secretome was enriched with proteins related to hydrolase, oxidoreductase, aspartic-type endopeptidase activities, as well as transferase activity, among others (Figure 4D).

### 2.4. T. atroviride and Arabidopsis Secretomes Were Enriched with a Plethora of Enzymes

Furthermore, the enzymatic functions of *Arabidopsis* and *T. atroviride* proteins were predicted and classified. According to this analysis, 539 proteins (69%) of *T. atroviride* and 58 (49%) of *Arabidopsis* were classified as enzymes (Appendix A). As shown in Figure 5, the main sub-subclasses of enzymes modulated at the different times in the *T. atroviride* secretome were putative glycosidases, carboxylic ester hydrolases, enzymes with NAD^+^ or NADP^+^ as acceptor, hydro-lyases, and aspartic endopeptidases. Moreover, phosphoric monoester hydrolases and enzymes acting on linear amides were predicted at the three times of interaction (Figure 5A–C). Some sub-sub classes of enzymes were only modulated at a particular time point of interaction, such as enzymes acting on superoxide as an acceptor, which were exclusively identified at 24 h (Figure 6A), whereas transaminases were only found at 96 h (Figure 5C).

The most abundant enzymes in the *Arabidopsis* secretome at the three times of co-culture included putative peroxidases, chitinases, and cathepsins. Pectinesterases and lactoylglutathione lyases were also differentially modulated at the three times of interaction (Figure 6A–C). Two α-mannosidases and the enzyme GGAT1 were found only at 48 h (Figure 6B).

Sixty *Arabidopsis* proteins were predicted to lack enzymatic functions; therefore, they were classified into functional protein families (Appendix A). According to this analysis, these proteins belong to diverse families, including thioredoxin, jacalin, fasciclin, plastocyanin, and probable lipid transfer, among others (Appendix A). Of the 241 proteins from *T. atroviride* predicted without an enzymatic function, 138 proteins were classified into functional families such as eukaryotic Sm-like (LSM), fungal hydrophobin, glucosyltransferase, and glycosyl hydrolase among others (Appendix A).

In *T. atroviride*, the glycosidase sub-subclass was over-represented by enzymes likely involved in the degradation of plant cell wall polysaccharides, such as cellulases, endo-1,4-β-xylanases, xyloglucan-specific, endo-β-1,4-glucanases, cellulose 1,4-β-cellobiosidases, mannan endo-1,6-α-mannosidases, and β-glucosidases (Table 1 and Appendix A). Furthermore, some carboxylic ester hydrolases with a potential role in cell wall degradation were identified, including a cutinase, an acetylxylan esterase, and a lysophospholipase. Conversely, other enzymes involved in the hydrolysis of plant polysaccharides decreased during the time-course of interaction, such as β-glucosidases, xylan 1,4-β-xylosidases, and α-L-arabinofuranosides (Table 1 and Appendix A).

Additionally, in the *T. atroviride* secretome, a group of antioxidant enzymes involved in ROS detoxification was significantly increased in response to the plant, such as those acting on a peroxide as acceptor. Some of these were exclusively increased at 24 h of interaction (Table 1 and Appendix A).

Furthermore, in the *T. atroviride* secretome, phosphoric monoester hydrolases that participate in protein dephosphorylation and nucleotide catabolic processes were increased by the presence of the plant, including a putative 2-phosphoglycolate phosphatase, an acid phosphatase, and a 3-phytase, among others (Table 1 and Appendix A).

Among the peptidases modulated at different times in the *T. atroviride* secretome, the aspartic endopeptidases were the most over-represented (Figure 5A–C). Serine-type carboxypeptidases and a metallocarboxypeptidase were accumulated mainly at 24 h of co-culture (Figure 5A), whereas at 96 h, an accumulation of several dipeptidyl/tripeptidyl peptidases was observed (Figure 5C and Appendix A).

A group of enzymes involved in oxidative stress response was prominent among the increased subsets of *Arabidopsis* proteins, including the thioredoxins (TRXs) BAS1A, TRXH3, TRX5, TRXM1, and ATHM2 (Appendix A). Other enzymes with a known role in oxidative stress response comprise two superoxide dismutases, SODC2 and FSD1 (Fe superoxide dismutase 1) (Appendix A). Furthermore, eight class III peroxidases (PER3, PER22, PER32, PER34, PER39, PER52, PER69, and PER71) were differentially accumulated in the *Arabidopsis* secretome (Table 2 and Appendix A).

A group of enzymes, which may be involved in plant defense, was increased in the plant secretome, including the hevein-like HEVL and the plant defensin PDF1.2. The decreased enzymes included the glucan endo-1,3-β-d-glucosidase PR5, PR1, and two defensin-like proteins (DF195 and DF206) (Appendix A). In addition, nine putative chitinases were found, most of which were decreased mainly at 48 and 96 h. Only the chitinase F15K9.17 increased in the presence of the fungus (Table 2 and Appendix A).

The accumulation of the serine protease inhibitor UPI, the Kunitz trypsin inhibitor KTI1, and the Kunitz trypsin inhibitor KTI4, which are linked to plant defense, was observed in the *Arabidopsis* secretome. In contrast, the serine protease inhibitor (At2g38870) and the Kunitz trypsin inhibitor KTI5 were strongly decreased through all interaction times (Appendix A). Other abundant proteins in the plant secretome were cysteine endopeptidases with a known role in plant defense, and several putative cathepsins (Table 2 and Appendix A) of the papain family of cysteine proteases (Appendix A). Other peptidases in the *Arabidopsis* secretome included a lysosomal Pro-Xaa carboxypeptidase and two subtilisin-like proteases (Appendix A). Furthermore, our plant secretome data revealed an enrichment of enzymes with no known roles in symbiotic interactions, such as phosphoglycolate phosphatase, PGP1B, ribulose bisphosphate carboxylase, RBCL (large chain), and GGAT1, which are related to photorespiration (Table 2 and Appendix A).

### 2.5. GGAT1 Plays a Minor Role in Plant Growth Stimulation by T. atroviride

As mentioned above, we found increased levels of GGAT1 in the plant secretome. It was reported that GGAT1 is secreted in *Arabidopsis* extracellular vesicles [28]. Abundance in the secretome decreased in response to the bacterial phytopathogen *P. syringae* (Appendix A, [28]), which indicates a potential role in plant–microbe interaction. However, its role in plant–pathogen or plant–beneficial microbe interaction has not been studied. Based on this information, we hypothesized that GGAT1 could be potentially involved in *Arabidopsis*–*T. atroviride* interaction. Thus, to test whether GGAT1 is required in *Arabidopsis* for plant growth stimulation by *T. atroviride*, the plant growth phenotypes and dry weights of Col-0 and *ggat1-2* lines were determined. As shown in Figure 7A,B, no significant differences in plant growth phenotype between *ggat1-2* and Col-0 control plants were observed. Both Col-0 and *ggat1-2* plants treated with *T. atroviride* showed significantly enhanced growth at 21-post treatment compared with untreated seedlings. However, the plant growth stimulation of *ggat1-2* by *T. atroviride* was significantly smaller (*p* < 0.05) than wild type Col-0 seedlings under the same conditions. This result indicated that GGAT1 is partially required for plant growth stimulation by the beneficial fungus.

### 2.6. GGAT1 Is Potentially Involved in the Negative Regulation of the Systemic Resistance against B. cinerea

Furthermore, we analyzed whether GGAT1 is required for the induction of plant systemic resistance by *T. atroviride* against *B. cinerea*. To this end, *ggat1-2* plants previously treated or not with *T. atroviride* were infected with *B. cinerea,* and the leaves damage was evaluated. *ggat1-2* seedlings untreated with *T. atroviride* exhibited decreased leaves damage compared to Col-0 untreated plants with *T. atroviride* (Figure 7C,D). *ggat1-2* plants pretreated with *T. atroviride* and post infected with *B. cinerea* showed slightly more foliar damage area compared to Col-0 seedlings pretreated with the mutualistic fungus. The differences were statistically significant. These results suggest that GGAT1 participates in the negative regulation of the systemic resistance against *B. cinerea* and that this enzyme is partially required for the induction of plant systemic resistance by *T. atroviride* against the phytopathogen.

### 2.7. In Arabidopsis, GGAT1 Participates in the Resistance against B. cinerea, Tentatively through a Mechanism Involving H_2_O_2_ Production

Previously, it was reported that an insertional mutant allele of *GGAT1* (*ggt1-1*) in *Arabidopsis* shows increased levels of hydrogen peroxide (H_2_O_2_) in leaves [34]. Therefore, we hypothesized that in *Arabidopsis*, GGAT1 could be involved in the response to both *T. atroviride* and *B. cinerea*. To test whether *ggat1-2* allele is affected in the production of H_2_O_2_, Col-0, and *ggat1-2*, leaves of control plants were stained with 3,3′-diaminobenzidine (DAB). Indeed, notable precipitates of DAB were visible in the leaves of *ggat1-2* plants compared with Col-0 (Figure 8), whereas after 24 h of challenging against *B. cinerea*, DAB precipitates in *ggat1-2* leaves were less visible than Col-0, indicating that the systemic production of H_2_O_2_ in *ggat1-2* leaves is negatively affected upon *B. cinerea* attack (Figure 8). Furthermore, DAB staining revealed that *T. atroviride* triggered the accumulation of H_2_O_2_ in Col-0 leaves, whereas low precipitate of DAB was observed in leaves of *ggat1-2* plants pretreated with *T. atroviride* (Figure 8). Moreover, *ggat1-2* plants pretreated with *T. atroviride* and infected with *B. cinerea* showed low levels of H_2_O_2_ accumulation compared with Col-0 under the same conditions (Figure 8).

## 3. Discussion

### 3.1. Arabidopsis and T. atroviride Secreted a Diverse Array of Proteins during Their Interaction

The proteins secreted by the host plant and its fungal partner may play key roles in the establishment of a beneficial relationship. In this work, we identified a total of 126 secreted proteins of *Arabidopsis* and 1027 of *T. atroviride* over a time course in a semi-hydroponic system. In a previous study [26], it was reported that a total of 43 proteins of *T. virens* and 95 of maize are secreted during their interaction under hydroponic conditions. *Here*, we show that seven out of 97 proteins of maize overlapped with their *orthologues* in *Arabidopsis*. These included the PVR3-like protein, the chitinase B1 (C0P451_MAIZE), the peroxidase PER67 (A0A1D6QGI0_MAIZE), the Barwin superfamily protein WIN1 (B6SH12_MAIZE), the endochitinase A (B4FTS6_MAIZE), the osmotin-like protein OSM34 (A0A1D6GKZ3_MAIZE), and a lactoylgluthatione lyase (C0PK05_MAIZE). It was also reported that *Arabidopsis* secretes three endochitinases (At2g43570, At2g43620, and At3g12500) and the osmotin-like protein OSM34 (At4g11650) in response to the mutualistic fungus *Piriformospora indica* [35]. These findings suggest that *Arabidopsis* and maize plants respond to beneficial fungi by secreting a common array of enzymes related to plant defense. Moreover, a total of 280 secreted proteins of *T. virens* growing alone or in co-culture with maize in a hydroponic system were identified [27]. The comparison of our dataset with that reported by [27] showed that 159 proteins of *T. atroviride* overlapped with their orthologous proteins in *T. virens*; most of such proteins were glycosidases and peptidases. It is tempting to speculate that glycosidases of *Trichoderma* spp. could be playing a role in plant–*Trichoderma* interaction. In support of this, the silencing of the *thpg1* gene that encodes the endopolygalacturonase ThPG1 in *T. harzianum* T34 showed reduced tomato root colonization and diminished growth in minimal medium supplemented with pectin as carbon source [36]. These data together indicate that *T. atroviride*, *T. virens*, and tentatively other *Trichoderma* species respond to their host plant by secreting a common subset of lytic enzymes, mainly those related with the breakdown of cell wall components to colonize the plant roots.

### 3.2. Arabidopsis and T. atroviride Secrete Proteins through Conventional and Non-Conventional Secretion Pathways

Among the differentially modulated proteins in both secretomes, 71% of the *Arabidopsis* proteins and 40% of *T. atroviride* contain a putative N-terminal SP, which is a secretion mark to release proteins to the medium, or for localization into or across the cell membrane. In eukaryotic cells, proteins with an SP are directed to the translocation apparatus of the ER and then, through vesicular transport from the ER to the Golgi, to be secreted outside of the cell [37]. This form of conventional secretion has been suggested for proteins secreted by maize plants in response to *T. virens* [26]. These findings suggest that during their interaction, *Arabidopsis* and *T. atroviride* deliver proteins to the extracellular milieu through conventional secretion systems. The putative mechanisms involving extracellular enzymes in the *Arabidopsis–Trichoderma* interaction are shown in our hypothetical model (Figure 9).

According to cellular component analysis, several *Arabidopsis* proteins were predicted to have an intracellular origin, such as JAL30 (jacalin-related lectin 30), RBL, GGAT1, PGP1B, FSD1 (Fe superoxide dismutase 1), and RCA (ribulose bisphosphate carboxylase/oxygenase activase) (Table 2). Interestingly, all these proteins were found in endomembrane compartments [38], and in EVs of *Arabidopsis* [28]. According to our analysis, all these proteins lack an SP and were potentially secreted by an unconventional secretion system (Appendix A and Figure 9).

Furthermore, our analysis revealed that in the *T. atroviride* secretome, 43% of the proteins lack an SP. Similar results were reported for *T. virens* proteins secreted in response to maize plants [26]. In some fungi, extracellular proteins that do not contain an SP are secreted by unconventional secretory pathways [39]. For example, the filamentous fungus *Magnaporthe oryzae* delivers cytoplasmic effector proteins into the host plant cell through a form of secretion that involves exocyst components and the Sso1 *t*-SNARE protein. In this form of secretion, the *M. oryzae* effector proteins are accumulated in a plant-derived membrane-rich interfacial structure called biotrophic interfacial complex and are ultimately delivered into the host cell [40]. Exocytic SNARE proteins have been reported to localize at subapical hyphae in *T. reesei*, which suggests the existence of an exocytic pathway for protein secretion in this fungus [41]. Interestingly, we found that a putative vesicle transport *t*-SNARE protein (JGI id: 146306) was accumulated in *T. atroviride* secretome at 96 h of co-culture with *Arabidopsis* (Appendix A).

Additionally, some of the proteins identified in the secretome of *T. atroviride* may be released to the extracellular milieu through EVs. This idea is reinforced by a previous proteomic analysis of vesicles secreted by *T. reesei* during its growth in the presence of cellulose as a carbon source [42]. These authors reported the presence of 188 proteins inside vesicles, including glycosidases, peptidases, and chaperones. Here, we analyzed all the proteins reported by [42] to infer whether they lack an SP. Indeed, by using SignalP, we predicted that 177 out of 188 reported proteins (94%) lack an SP (data not shown). All these findings together with our results suggest that unconventional secretion systems may be a mechanism by which *Trichoderma* spp. deliver proteins to the extracellular milieu to establish a crosstalk with their host plants.

### 3.3. T. atroviride Secretes Enzymes Involved in Oxidative Stress Response, Cell Wall Degradation, Hydrolysis of Organic Phosphate Sources, and Peptidases during Its Interaction with Arabidopsis

In the *T. atroviride* secretome, we found several proteins potentially involved in oxidative stress response, which are represented mainly by enzymes acting on a peroxide as an acceptor, including catalases and catalase-peroxidases. Catalase-peroxidases possess both catalase and peroxidase activity, and they are thought to have a cell protective functions under oxidative stress [43]. In phytopathogenic fungi, catalases and peroxidases play important roles in defense against oxidative stress. For instance, the catalase-peroxidase CPXB secreted by *M. oryzae* plays a pivotal role in fungal defense against H_2_O_2_ accumulation in the epidermal cells of rice [44]. The enrichment of antioxidant enzymes in the *T. atroviride* secretome suggests a response of the fungus to cope with ROS produced by *Arabidopsis*, as shown for leaves in Figure 8.

The *T. atroviride* secretome analysis revealed the accumulation of many glycosidases, mainly during the early steps of interaction with *Arabidopsis*. These enzymes include cellulases, cellulose 1,4-β-cellobiosidases, and β-glucosidases (Table 1 and Appendix A), which could be potentially involved in cell wall degradation of its host. Cellulases hydrolyze β-1,4-d-glucan bonds of cellulose to produce cellobiose and other short oligosaccharides, which are hydrolyzed to glucose by β-glucosidases [45,46]. Other polysaccharide-hydrolyzing enzymes were increased in the *T. atroviride* secretome, including xyloglucan-specific, endo-β-1,4-glucanases, and endo-1,4-β-xylanases, which hydrolyze the cell wall polysaccharides, xyloglucan, and xylan, respectively. In addition, other putative CWDEs were increased in *T. atroviride* secretome, such as a group of carboxylic ester hydrolases, an acetyl xylan esterase, and a cutinase. Acetyl xylan esterases remove side-chain residues from xylan backbones [47], whereas cutinases hydrolyze ester bonds of the plant polymer cutin [48]. In agreement with our results, an accumulation of extracellular glycosidases was reported in the *T. virens* secretome during its coculture with maize plants [26,27]. This supports the hypothesis that *T. atroviride* secretes CWDEs to disrupt the plant cell wall to gain access into the root epidermis and cortex to establish a mutualistic relationship (Figure 9).

The analysis of *T. atroviride* secretome revealed the presence of several enzymes involved in the hydrolysis of phosphate compounds, which was represented mainly by phosphoric monoester hydrolases. Phosphoric monoester hydrolases secreted by microbes can hydrolyze organic phosphate sources from the soil, such as inositol phosphates (phytates), nucleic acids, sugar phosphates, and phospholipids [49]. Reactions catalyzed by phosphoric monoester hydrolases such as phytases, acid phosphatases, and inositol-phosphate phosphatases release inorganic phosphorous (P_i_), which is taken up by plants and microorganisms for their metabolism [50]. Several studies have shown the capability of *Trichoderma* spp. to solubilize phosphate in vitro [51,52,53] and under stress conditions [54,55,56]. *Phytases* are phosphatases that hydrolyze phytic acid preferentially into inositol and P_i_ [50], whereas acid phosphatases hydrolyze phosphomonoester and amides substrates, thereby transforming organic phosphate into a soluble inorganic form [57]. Phytase activity has been observed in several beneficial fungi, including *Rhizopus microsporus* [58], *Funneliformis mosseae, Claroideoglomus etunicatum* [49], and *Trichoderma* spp. [51]. We hypothesize that during *T. atroviride* interaction with its host plant, it secretes phosphoric monoester hydrolases to transform organic phosphate sources into a soluble inorganic form, which can be taken up by the fungus and the host plant for their metabolism (Figure 9).

The *T. atroviride* secretome analysis showed an array of peptidases, some of which were increased during its interaction with the plant. Peptidases have been studied mainly in phytopathogenic fungi. Plant pathogens secrete proteases that modify or degrade PR proteins with antimicrobial activity produced by the host plant, including chitinases, β-1,3-glucanases, and proteases [59]. For instance, *Verticillium dahliae* secretes the Ser protease 1 (VdSSEP1) into the apoplast of cotton, where it hydrolyzes the extracellular chitinase Chi28 of the plant [60]. The Zn-metalloproteinase Fv-cmp from *Fusarium verticillioides* cleaves the class IV chitinases of the plant, which are plant defense proteins that bind and degrade the chitin of the fungal cell wall [61]. In the symbiotic fungi *Oidiodendron maius*, *Rhizoscyphus ericae*, and *Meliniomyces bicolor*, genes encoding proteases such as aspartic proteases, glutamic proteases, and subtilisins are upregulated during their interaction with their host plant [62]. Probably, *T. atroviride* secretes peptidases to inactivate the action of plant defense proteins or to obtain nitrogen sources from the degraded peptides for its growth (Figure 9).

### 3.4. Arabidopsis Responds to the Presence of T. atroviride by Secreting Enzymes Involved in ROS Generation, Oxidative Stress Response, Defense Response, and Photorespiration

Several peroxidases with a known role in ROS generation were increased in the *Arabidopsis* secretome during its interaction with *T. atroviride*. In response to invading microbes, plants accumulate ROS in the apoplast, including H_2_O_2_, hydroxyl radicals (•OH), superoxide radicals (O_2_•^−^), and nitric oxide (NO), among others, leading to the so-called “oxidative burst” [63]. In *Arabidopsis*, the extracellular oxidative burst triggered by MAMPs involves the class III secretory peroxidases PRX33 and PRX34 [64,65]. In this regard, the H_2_O_2_ generated in the apoplast by PRX33 and PRX34 increases the colonization success of the phytopathogen *Alternaria brassicicola* [66]. Mechanisms involving H_2_O_2_ production have been described during plant–symbiotic fungi associations. For example, maize roots treated with the mycorrhizal fungus *Glomus intraradices* accumulate H_2_O_2_, probably to regulate the proliferation of the fungus within the plant roots [63]. Together, these data suggest that during its interaction with *T. atroviride*, the plant activates its ROS-producing systems as a mechanism to control the fungus proliferation within its roots (Figure 9).

Conversely, some peroxidases identified in the plant secretome, including PRX52, PRX69, and PRX71, were decreased through three times the interaction with *T. atroviride*. In agreement with these findings, a group of peroxidases are reduced in the root secretome of maize plants in response to the colonization by *T. virens* that correlates with a reduction in peroxidase activity, suggesting that *T. virens* manipulates its host stress oxidative response [26]. It is tempting to hypothesize that *T. atroviride* manipulates the *Arabidopsis* stress oxidative response as proposed for *T. virens* in maize seedlings.

A set of TRXs involved in oxidative stress response was increased in the *Arabidopsis* secretome in response to *T.* atroviride (Table 2 and Appendix A). Some TRXs are secreted into the apoplast in response to pathogen attack or in response to abiotic stress, where they function as antioxidants, facilitating the reduction of other proteins [67,68]. TRXs play a major role in maintaining the cell in a reducing environment, interacting with target proteins to control their functions [69]. During their interaction with target proteins, TRXs reduce the disulfide bridges formed between cysteine residues using their highly conserved thiol groups [68,70]. It is tempting to speculate that during *T. atroviride* interaction with *Arabidopsis*, the plant secretes TRXs to regulate the activity of cysteine-rich extracellular proteins secreted by both organisms to maintain the redox homeostasis (Figure 9).

Plants have to fight against pathogens through the secretion of protease inhibitors that act against proteolytic enzymes liberated by the pathogen, including CWDEs [71,72]. According to the *Arabidopsis* secretome analysis, some protease inhibitors increased in response to *T. atroviride*, including the unusual serine protease inhibitor UPI, which is involved in plant defense against necrotrophic fungi [73] and the Kunitz trypsin inhibitor KTI1 involved in modulating programmed cell death (PCD) in plant–pathogen interactions and in defense against *P. syringae* [71]. Furthermore, Kunitz inhibitors can inhibit multiple types of hydrolytic enzymes, such as serine and cysteine proteases [72]. The secretion of protease inhibitors by *Arabidopsis* during its interaction with *T. atroviride* suggests that they could be working in the apoplast, inhibiting the activity of extracellular proteolytic enzymes of the fungus and helping to prevent the cell wall damage (Figure 9).

In *Arabidopsis*, a set of PR proteins with antimicrobial activity is secreted into the apoplast in response to pathogens and mutualistic fungi [35,74]. Here, several PR proteins increased in *Arabidopsis* secretome mainly at 24 and 48 h of interaction with *T. atroviride*, but not at later times. Overall, these findings suggest that *T. atroviride* is perceived as a pathogen by *Arabidopsis* at the first steps of the interaction, but it also suggests the suppression of plant immunity by the fungus at later times (Figure 9).

Here, we identified some enzymes related with photorespiration (Table 2 and Appendix A). In the photorespiratory pathway, the key CO_2_ fixation enzyme ribulose bisphosphate carboxylase (RuBisCo) synthesizes the metabolite 2-phosphoglycolate (2-PG). Accumulation of higher levels of 2-PG in the cell causes several negative effects in the plant, such as the inhibition of the Calvin cycle enzyme triosephosphate isomerase [75]. The degradation of 2-PG requires the combined action of several enzymes of the photorespiratory pathway, including 2-phosphoglycolate phosphatases (PGLPs), glycolate oxidases (GOXs), and glutamate:glyoxylate aminotransferases (GGATs) [75]. Interestingly, we identified two enzymes related with the degradation of 2-PG, PGP1B, which dephosphorylates 2-PG to produce glycolate [76], and GGAT1, that catalyzes the conversion of glyoxylate to glycine during the photorespiration [77]. The degradation of glyoxylate is crucial, because its accumulation interferes with photosynthesis through the inhibition of RuBisCo [78]. We found that PGP1B was increased at the three times of interaction with the fungus, whereas GGAT1 was only accumulated at 48 h of co-culture (Table 2 and Appendix A).

Our analysis indicated that RBCL was secreted by *Arabidopsis* when it was grown under control conditions (Appendix A). However, this enzyme was decreased at 24 and 48 h of co-culture with the fungus (Table 2 and Appendix A). We hypothesize that during the early steps of the interaction, the plant secretes into the apoplast enzymes related to photorespiration, including PGP1B and GGAT1, which causes the accumulation of glyoxylate, interfering with the activity of enzymes related with photosynthesis, including RuBisCo (Figure 9). This negative effect on photosynthesis is probably alleviated at late stages of the interaction when the beneficial relationship between both organisms is established. In this regard, some *Trichoderma* strains enhance the photosynthetic rate in their host plants [79,80]. Furthermore, photosynthetic enzymes were differentially accumulated in the cotyledons and shoots of cucumber and maize plants after root colonization by *Trichoderma asperellum* strain T34 and *Trichoderma harzianum* strain T22, respectively [81,82]. Since GGAT1, PGP1B, and RBCL are related with the photorespiration in photosynthetic tissues (i.e., in leaves), the identification of this enzymes in the plant root secretome was particularly interesting.

### 3.5. GGAT1 Is Partially Required for Plant Growth Stimulation by T. atroviride

In this study, we characterized a mutant allele of *GGAT1* (*ggat1-2*) in *Arabidopsis* to test whether its product plays a role in plant growth stimulation by *T. atroviride*. Under our experimental conditions, *ggat1-2* exhibited no apparent differences in the growth and development compared with Col-0 plants; however, it was reported that a different mutant allele of *GGAT1* (*ggt1-1*) exhibited slight or no differences in growth phenotype compared with Col-0 plants under high-CO_2_ conditions [77,83]. Nevertheless, our results of plant–*Trichoderma* interaction showed that GGAT1 is partially required for plant growth stimulation by *T. atroviride*, but this phenotype is independent of root colonization by the fungus since similar results of CFU were determined for Col-0 and *ggat1-2* seedlings (Appendix A).

### 3.6. GGAT1 Negatively Regulates the Plant Systemic Resistance against B. cinerea Tentatively through a Mechanism that Involves Altered H_2_O_2_ Production

We found that *ggat1-2* leaves of untreated seedlings with *T. atroviride* exhibited decreased damage provoked by *B. cinerea* compared to Col-0 plants. Contrastingly, *ggat1-2* plants pretreated with *T. atroviride* showed a significant increase in lesion area than those of the Col-0 (Figure 7). These findings suggest that GGAT1 participates negatively in the resistance against *B. cinerea* and plays a minor role in inducing the plant systemic resistance by *T. atroviride* against the phytopathogen.

Previously, it was reported that a mutant allele of *GGAT1* (*ggt1-1*) in *Arabidopsis* shows increased levels of H_2_O_2_ in leaves, which is consistent with the role of GGAT1 in photorespiration [34]. Indeed, we confirmed that the absence of *GGAT1* in *Arabidopsis* causes a constitutive accumulation of H_2_O_2_ in leaves (Figure 8). GGAT1 is localized in peroxisomes [84]. Photorespiratory metabolism in peroxisomes produces H_2_O_2_ through the oxidation of glycolate to glyoxylate by GOXs [85]. However, the high production of H_2_O_2_ in *ggat1-2* may be due to an indirect effect by the absence of GGAT1, since this protein is not directly involved in H_2_O_2_ production in the photorespiratory pathway [75]. This increased production of H_2_O_2_ in *ggat1-2* may be confined to photosynthetic tissues as suggested [34], whereas it would not be expected in roots. Consistent with this hypothesis, *GGAT1* transcripts were detected in mature leaves and in green siliques of *Arabidopsis* but not in roots [84]. We hypothesized that the altered production of H_2_O_2_ in *ggat1-2* is linked with the enhanced resistance to *B. cinerea* observed in this mutant (Figure 9). Consistently with this hypothesis, high levels of H_2_O_2_ in tomato leaves are related to enhanced resistance to *B. cinerea* [86].

Furthermore, in addition to the altered levels of H_2_O_2_ in *ggt1-1*, decreased levels of abscisic acid (ABA) are observed under polyethylene glycol (low water potential) or NaCl stresses [34]. In addition, it is well known that ABA accumulation antagonizes JA-ET signaling pathways [87]. Furthermore, ABA deficient mutants results in the upregulation of JA-ET responsive genes and correlates with enhanced resistance in *Arabidopsis* to the necrotrophic pathogen *Fusarium oxysporum* [87]. Based on these data, we hypothesize that increased levels of H_2_O_2_ in *ggat1-2* plants could affect ABA accumulation as reported [34], leading to the upregulation of JA-ET responsive genes and triggering an enhanced resistance against *B. cinerea*.

In agreement with a previous report [88], DAB staining revealed that Col-0 seedlings co-cultivated with *T. atroviride* accumulate H_2_O_2_ in leaves. Furthermore, *B. cinerea* triggers the systemic accumulation of H_2_O_2_ in Col-0 leaves. Conversely, *ggat1-2* exhibited reduced H_2_O_2_ levels in response to *B. cinerea* or *T. atroviride* compared to Col-0 under the same conditions (Figure 8), indicating that the production of H_2_O_2_ in the *ggat1-2* mutant is negatively affected in response to the pathogen or the beneficial fungus. All these observations suggest that GGAT1 plays a role in regulating plant response to beneficial fungi and necrotrophic pathogens tentatively through a mechanism involved in H_2_O_2_ production.

Concluding, our quantitative proteomics and bioinformatic analysis of proteins secreted by *Arabidopsis* and *T. atroviride* indicate that during their interaction, both organisms secrete an array of proteins with enzymatic and non-enzymatic functions. *T. atroviride* secretes mainly glycosidases, aspartic endopeptidases, and dehydrogenases. In response to the presence of the fungus, the plant secretes peroxidases, cysteine endopeptidases, thioredoxins, and enzymes related to catabolism of secondary metabolites. Additionally, *Arabidopsis* secretes the photorespiratory enzyme glutamate:glyoxylate aminotransferase, GGAT1, which is partially required for plant growth stimulation and the induction of systemic resistance against *Botrytis* by *T. atroviride*. Finally, GGAT1 participates in the regulation of plant systemic resistance against *B. cinerea* tentatively through a mechanism involving H_2_O_2_ production.

## 4. Materials and Methods

### 4.1. Plant and Fungal Growth Conditions for In Vitro Experiments

*Arabidopsis thaliana* Columbia (Col-0) ecotype was used as the wt plant through this work. Seeds of *ggat1-2* T-DNA insertional mutant (SALK_064982C) were obtained from the *Arabidopsis* Biological Resource Center (ABRC). *ggat1-2* homozygous T-DNA insertional mutants were genotyped (Appendix A) using the primers listed in Appendix A. For in vitro experiments, *A. thaliana* Col-0 seeds were sterilized by dry sterilization using 50 mL of commercial bleach (3.5%) with 1 mL of hydrochloric acid (32%) during four h in a closed chamber inside a fume hood. Sterilized seeds were stratified in sterile Petri dishes containing Murashige–Skoog (MS) medium (0.5× Murashige–Skoog medium, Duchefa Biochemie, Harleem, Netherlands), 2.5 mM MES buffer (Merck, Darmstadt, Germany) and 0.8%(*w*/*v*) plant agar (Duchefa Biochemie, Harleem, Netherlands), pH = 5.7) in the dark at 4 °C for 2 days. Seeds were germinated on MS agar plate at 25 °C under 16/8h light/dark cycles (130 μmol m^−2^s^−1^) for two days. *Trichoderma atroviride* IMI 206040 and *Botrytis cinerea* B05.10 [89] were routinely grown at 28 °C on potato dextrose agar (PDA) (DIFCO) for 7 and 14 days, respectively. *T. atroviride* conidia were collected with sterile distilled water and adjusted at 1 × 10^6^ mL^−1^, whereas *B. cinerea* conidia were adjusted to 5 × 10^5^ mL^−1^ in inoculation buffer (per 40 mL of stock solution: Sucrose 1.37 g, 1 M KH_2_PO_4_ 400 µL, 12.5% Tween20: 80 µL) [90].

### 4.2. Secretome System for Arabidopsis and T. atroviride Interaction Analysis

*Arabidopsis* and *T. atroviride* were grown in a semi-hydroponic system adapted from [27]. Twenty µL micropipette tip holders were cut for supporting the germinated seeds and filled with MS medium (0.8% agar). Thirty 2-day-old seedlings were transferred to the tip holder filled with MS medium and placed inside magenta boxes (Appendix A). Twenty-five mL of MS medium were added to each semi-hydroponic system, and every four days, the growing medium was replaced by fresh medium. For *T. atroviride* inoculum, fifty mL of potato dextrose broth (PDB) medium were inoculated with a suspension of *T. atroviride* (1 × 10^6^ conidia/mL) and incubated with continuous shaking (220 rpm) at 28 °C for 72 h. Actively growing mycelium was recovered by vacuum filtration through a 0.2 µm filter and resuspended in 25 mL of MS medium supplemented with 0.05% (*w*/*v*) sucrose. Three-week-old seedlings growing in the semi-hydroponic system box were inoculated with this fungal suspension. Uninoculated plants and *T. atroviride* were grown in separated semi-hydroponic systems as controls. The plants were maintained at 25 °C and 16/8 light/dark photoperiod with moderate shaking (60 rpm) on an orbital shaker by four weeks.

### 4.3. Secretome Samples Concentration

The culture medium from semi-hydroponic system boxes was filtered through monofilament nylon mesh followed by a 0.22 µm filter. Filtered secretomes were concentrated to a final volume of 500 µL using Amicon Ultra-15 centrifugal filter units (cutoff 3000 Da; Merck Millipore, Darmstadt, Germany) in a benchtop centrifuge at 4 °C. Then, the collected samples were treated with a protease inhibitor cocktail (Sigma, St. Louis, MO, USA). Total protein samples were verified for integrity by SDS-PAGE (15% acrylamide). Samples were stored at −80 °C until their analysis.

### 4.4. Identification of Secreted Proteins by Gel-Free Shotgun Proteomics

Fifty µL of each *Arabidopsis* and *Trichoderma* control samples or 10 µL of each interaction samples were used for the analysis. Shotgun protein identification and data analysis were performed according to Lamdan et al., 2015 [27]. The mass spectroscopy data from two biological replicates of each treatment at the different time conditions (24, 48 and 96 h) were analyzed using MaxQuant software 1.5.2.8 (www.maxquant.org, accessed on 1 October 2018) for peak picking identification and quantitation using the same software. The identifications were filtered for proteins identified with an FDR < 0.01. Complete proteomic data were deposited at the PRIDE database, with the accession number PXD023283.

### 4.5. Bioinformatic Analysis of Secreted Proteins

A comprehensive pipeline was performed to analyze secreted proteins and to identify their main features. The accession number of the identified proteins was uploaded at the UniProt database [91] and extracted in FASTA format with their UniProt ID. Sequences were submitted to Blast2Go [32] for Gene Ontology (GO) term analysis (cellular component, biological process, and molecular function). Blast2Go uses BLASTp [92] to find homologous proteins by comparing the submitted sequences against the non-redundant NCBI protein database. Subsequently, proteins were classified into functional families using HMMER v3.3.2 [93] and into enzymes according to the EC nomenclature system using BRENDA [33]. The presence of a signal peptide was predicted using SignalP v5.0 [94], and the presence of transmembrane helices was predicted using TMHMM v2.0 [95]. Proteins secreted by the classical pathway were predicted using SECRETOOL, which incorporates SignalP, and TMHMM for secretome prediction [29]. Prediction of non-classically secreted proteins was carried out using SecretomeP v2.0 [30] and OutCyte v1.0 [31].

### 4.6. Plant-Growth Promotion and Plant–Pathogen Challenge Assays

For plant growth promotion experiments, Col-0 and *ggat1-2* seeds were sown into plastic pots containing 1:1:3 perlite:vermiculite:peat moss as substrate, stratified at 4 °C for 2 days, and then transferred to a growth room at 23 °C under 12 h light/12 h dark regime. Six-day-old seedlings were transferred to 6 cm plastic pots containing substrate and allowed to grow as mentioned above. Ten days after their transfer, plants were root-inoculated with 500 µL of a suspension of 1 × 10^6^ conidia/mL of *T. atroviride* in 0.3× MS liquid medium. Plant inoculated in the roots with 500 μL of 0.3× MS were included as controls. Dry weight of 20 plants were determined individually at 21 post-treatments. Three independent experiments were performed.

*Arabidopsis thaliana* challenging experiments against *B. cinerea* were performed infecting three leaves of eight Col-0 or *ggat1-2* plants pre-treated or not (control plants) with *T. atroviride* (18-days-post treatment) using 10 µL of a suspension of 5 × 10^5^ conidia of *B. cinerea* in inoculation buffer (1 M KH_2_PO_4_, 12.5% Tween 20, 1% Sucrose). Plants inoculated with inoculation buffer were included as controls. Digital images were taken at 6 days post-infection. The lesion area was calculated as pixels per lesion using ImageJ software and converted to cm^2^ using as reference a ruler for each image. Three independent experiments were performed.

### 4.7. Detection of Hydrogen Peroxide (H_2_O_2_) in Arabidopsis Leaves Using 3,3-diaminobenzidine (DAB)

In situ H_2_O_2_ detection was performed using 12-day-old seedlings (Col-0 and *ggat1-2* lines) grown under control conditions using 3,3-diaminobenzidine (DAB) staining (Sigma-Aldrich) as previously reported [96]. Leaves were infected with 1 μL of a suspension of 5 × 10^5^ conidia of *B. cinerea* or with inoculation buffer (control). In parallel, Col-0 and *ggat1-2* seedlings pretreated or not with *T. atroviride* for 72 h were infected with *B. cinerea* or with inoculation buffer. Uninoculated leaves were detached after 24 h and infiltrated with DAB. Samples were observed under microscope at 4x magnification. Ten seedlings of each line were analyzed by treatment. Assays were repeated twice. Ten representative leaves of each condition were selected for DAB staining intensity determination using Fiji, an image processing package distributed by ImageJ software (https://imagej.net/Fiji, accessed on 29 May 2021).

### 4.8. Colonization Assay

Thirty 15-day-old Col-0 and *ggat1-2* plants grown on 0.5 X MS agar plate at 25 °C under 16/8 h light/dark cycles (130 μmol m^−2^s^−1^) were immersed in a *T. atroviride* 1 × 10^7^ conidia/mL suspension for 5 min. Col-0 and *ggat1-2* plants were placed on paper filters contained into Petri dishes moisturized with 3 mL of sterile distilled water. Then, plants were incubated 72 h as described above. Plants treated with sterile distilled water were included as control. Detached roots were weighed and immersed into 1% NaOCl (commercial bleach diluted with sterile distilled water) for 2 min. Then, roots were washed three times with sterile distilled water for 2 min each and homogenized using a mortar with 1 mL of PBS buffer (5 mM NaCl, 2.7 mM KCl, 10 mM Na_2_HPO_4_, 1.8 mM KH_2_PO_4_, pH 7.2). Two hundred µL of the homogenate were plated on PDA plates supplemented with 50 µg/mL of chloramphenicol and 0.5% Triton and the colony-forming units (CFU) were assessed at 72 h.

## Figures and Tables

**Figure 1 ijms-22-06804-f001:**
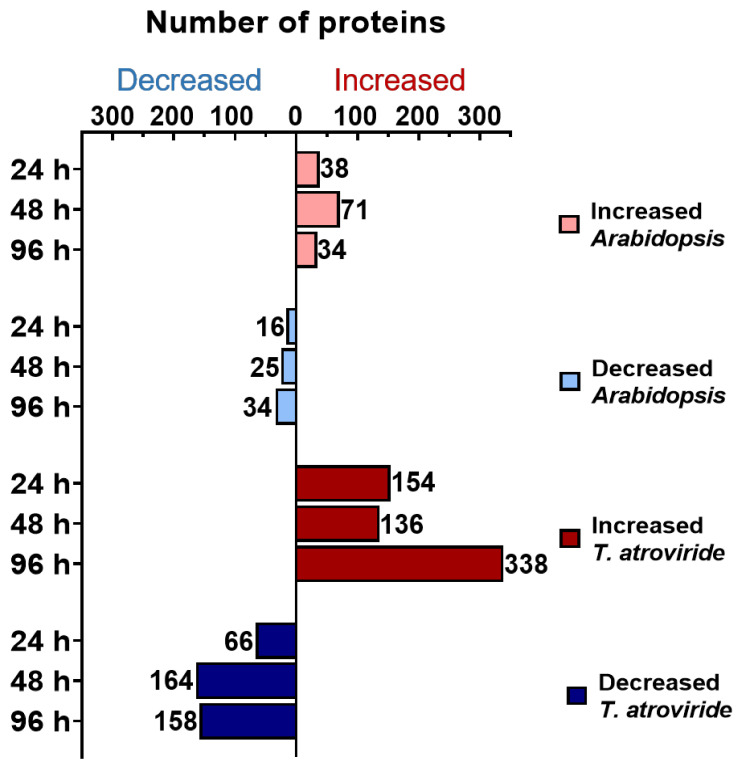
*Arabidopsis* and *T. atroviride* secreted proteins were differentially modulated mainly at 48 and 96 h of co-culture, respectively. Differential accumulation of *A. thaliana* and *T. atroviride* proteins during their interaction at 24, 48, and 96 h of co-culture. The *Arabidopsis* or *T. atroviride* proteins were classified as differentially expressed based on log_2_ ≥ 2.0 for increased and ≤−2.0 for decreased (with an FDR < 0.01) compared to their respective controls growing alone. Labels for each bar indicate the number of proteins.

**Figure 2 ijms-22-06804-f002:**
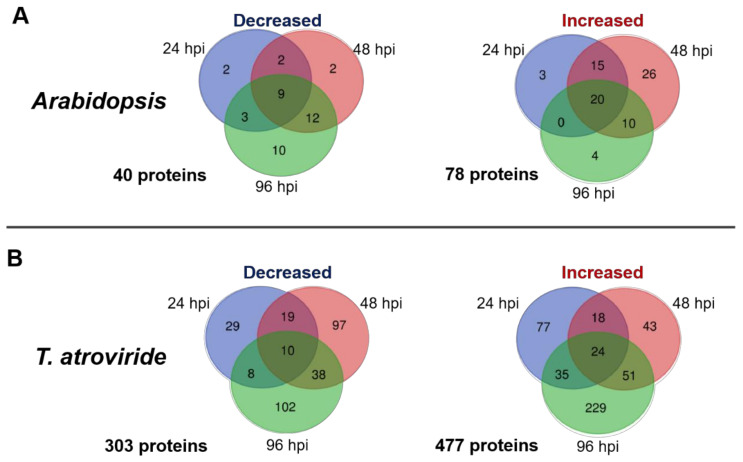
Venn diagrams showing the distribution of decreased and increased proteins at 24, 48, and 96 h of interaction. (**A**) *Arabidopsis* secretome. (**B**) *T. atroviride* secretome. Venn diagrams were constructed using the BEG tool (http://bioinformatics.psb.ugent.be/webtools/Venn/, accessed on 2 August 2020). The total number of differentially expressed proteins is depicted to the left and belove of each diagram.

**Figure 3 ijms-22-06804-f003:**
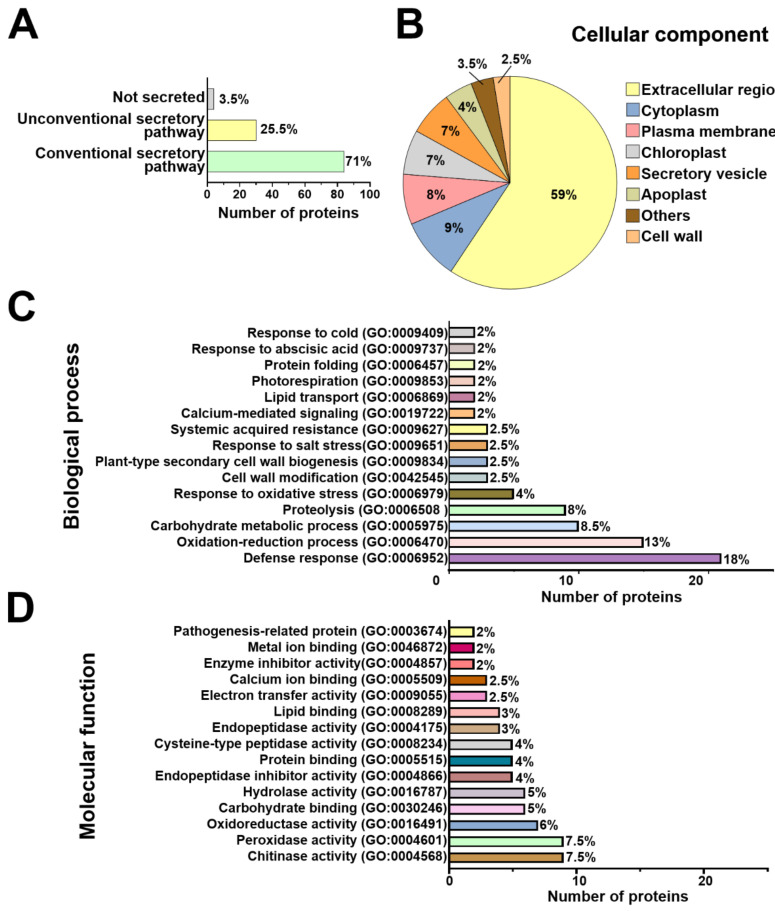
Prediction of putative secretion pathways and functional annotation of *Arabidopsis* secreted proteins. (**A**) Prediction of secretion pathways for *Arabidopsis* proteins during its interaction with *T. atroviride*. (**B**) Pie chart for the cellular component of *Arabidopsis* secreted proteins. Others include the vacuole, endoplasmic reticulum, and nucleus predicted. (**C**) Horizontal bar graphic for the top 15 biological processes of *Arabidopsis* secreted proteins predicted. Bars represent the number of proteins implied in a specific biological process. (**D**) Horizontal bar graphic for the top 15 molecular functions predicted. Prediction of secretion pathways was performed using SECRETOOL [29], SecretomeP v2.0 [30], and OutCyte v1.0 [31]. Functional annotation was performed with Gene Ontology (GO) terms using the Blast2GO software [32].

**Figure 4 ijms-22-06804-f004:**
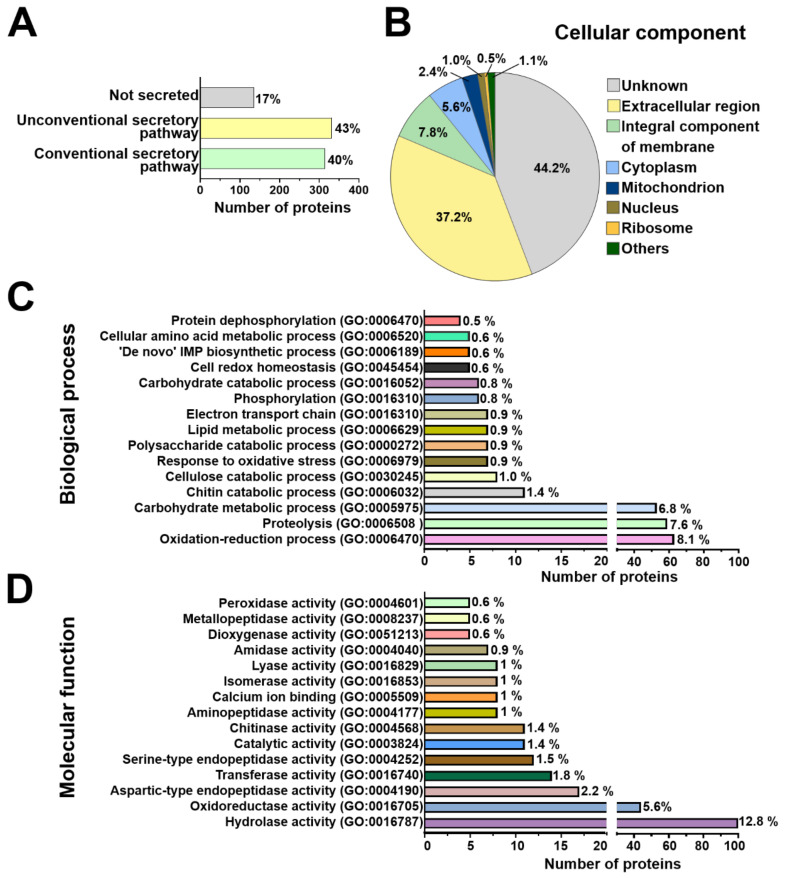
Prediction of putative secretion pathways and functional annotation of *T. atroviride* secreted proteins. (**A**) Prediction of secretion pathways of *T. atroviride* proteins during its interaction with *Arabidopsis*. (**B**) Pie chart for the cellular component of *T. atroviride* secreted proteins predicted by Blast2GO. Others included peroxisome, endosome, multivesicular body, cytoplasmic stress granule, cytoplasmic vesicle membrane, endoplasmic reticulum, and eukaryotic translation initiation factor 3 complex. (**C**) Horizontal bar graphic for the top 15 biological processes of *T. atroviride*-secreted proteins predicted by Blast2GO. Bars represent the number of proteins implied in a specific biological process. (**D**) Horizontal bar graphic for the top 15 molecular functions predicted by Blast2GO. The prediction of secretion pathways was performed using SECRETOOL [29], SecretomeP v2.0 [30], and the OutCyte v1.0 [31]. Functional annotation was performed with Gene Ontology (GO) terms using Blast2GO software [32].

**Figure 5 ijms-22-06804-f005:**
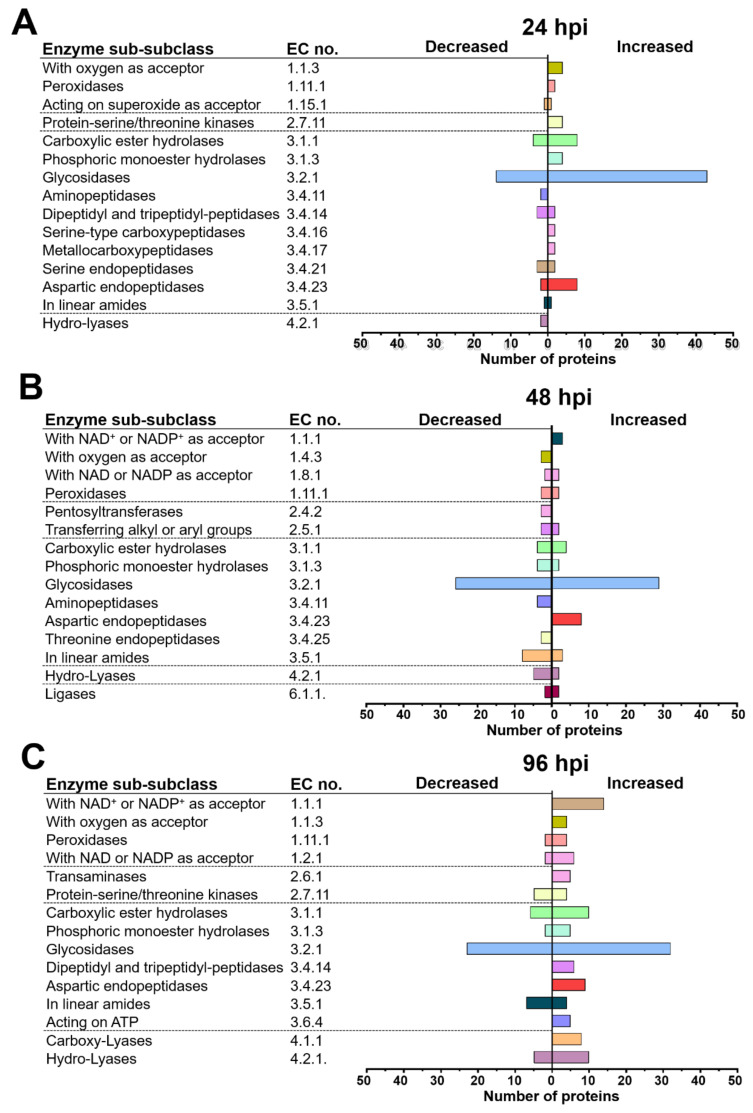
*T. atroviride* secreted mainly glycosidases and aspartic endopeptidases during its interaction with *Arabidopsis*. (**A**) Sub-subclasses of enzymes identified in the secretome of *T. atroviride* at 24 h, (**B**) 48 h, and (**C**) 96 h of interaction with *Arabidopsis.* The classification was based on the enzyme activity prediction using BRENDA [33].

**Figure 6 ijms-22-06804-f006:**
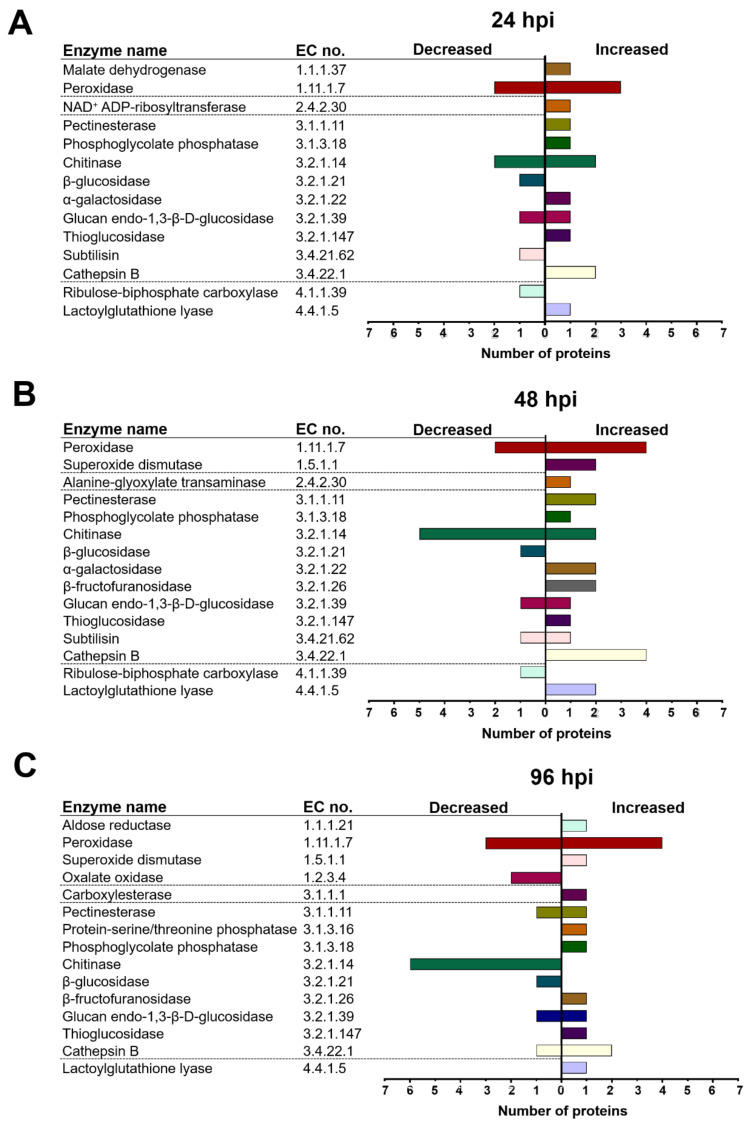
*Arabidopsis* responded to the presence of *T. atroviride* secreting mainly peroxidases and cathepsins B. (**A**) Enzymes identified in the *Arabidopsis*’s secretome at 24 h, (**B**) 48 h, and (**C**) 96 h of interaction with *T. atroviride.* The classification was based on the enzyme activity prediction using BRENDA [33].

**Figure 7 ijms-22-06804-f007:**
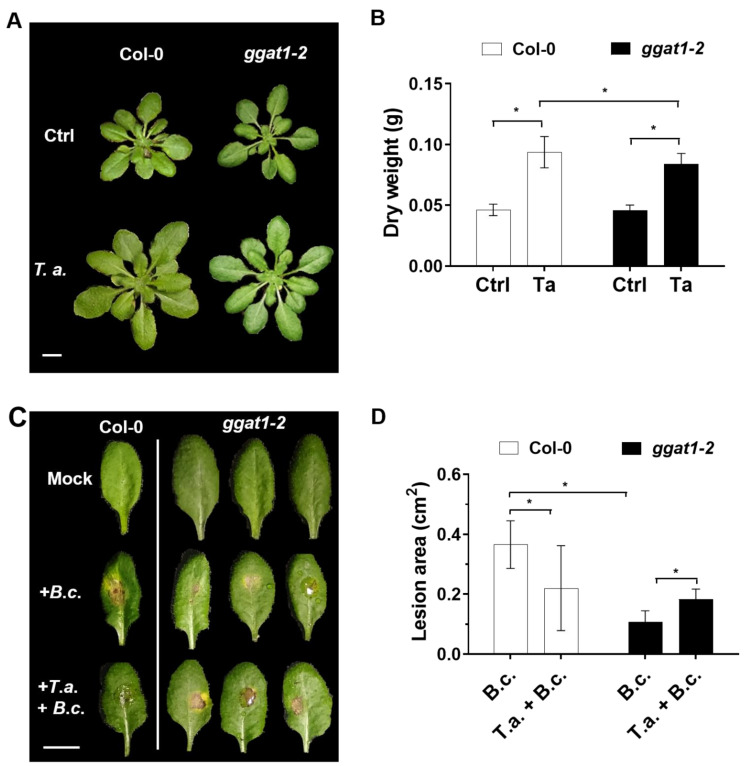
GGAT1 is partially required in *Arabidopsis* for plant growth induced by *T. atroviride* and potentially participates in the negative regulation of the systemic resistance against *B. cinerea*. Two-week-old *Arabidopsis* plants were root-pretreated with *T. atroviride*, whereas control plants were inoculated with buffer. (**A**) Plant phenotypes of growth response to *T. atroviride*. Pictures were taken at 18-days post-treatment. Scale bar, 1 cm. (**B**) Dry weights of individual plants were determined at 21 days post-treatment. The experiments were repeated thrice with similar results. The data represent the means of 20 plants. Results were analyzed using a Tukey multiple comparison test (α = 0.05). Asterisks indicate *p* < 0.05. (**C**) Comparison of leaves damage between Col-0 and *ggat1-2* seedlings inoculated with *B. cinerea* (+*B*. *c*). Plants treated with inoculation buffer were included as control (mock). Infections were carried out on three leaves from each plant previously treated or not with *T. atroviride* (18 days post-treatments, *T*. *a*. + *B*. *c*.). Pictures were taken at six days post infection. Scale bar, 1 cm. (**D**) Lesion sizes (cm^2^) of *Arabidopsis* leaves infected with *B. cinerea* were analyzed at six-days post infection (dpi) using ImageJ software. The experiments were repeated thrice with similar results. The data represent means of 24 leaves from a total of eight plants. Results were analyzed using a Tukey multiple comparison test (α = 0.05). Asterisks indicate *p* < 0.05. Error bars indicate SD.

**Figure 8 ijms-22-06804-f008:**
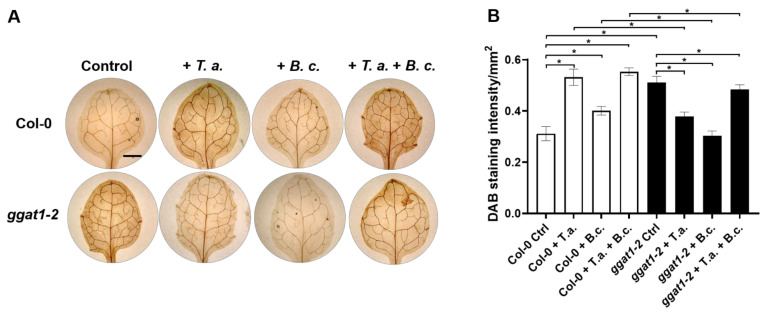
Histochemical detection of H_2_O_2_ in *Arabidopsis* leaves by DAB staining. (**A**) H_2_O_2_ in *Arabidopsis* leaves was visualized by DAB staining. Three leaves of each 12-day-old Col-0 or *ggat1-2* seedlings were inoculated with *B. cinerea* or with inoculation buffer as controls from plants previously treated (+*T. a*. + *B. c.*) or not with *T. atroviride* (+*B. c*.). The histochemical detection of H_2_O_2_ in uninoculated leaves was evaluated one-day post-inoculation with the phytopathogen. Leaves infiltrated with buffer were included as controls (not shown). Pictures were taken at 4x magnification in a microscope (ZEISS Primostar). Representative leaves from 10 plants are shown. The experiments were repeated twice with similar results. Scale bar, 1 mm. (**B**) DAB staining intensity as determined with Fiji, an image processing package distributed by ImageJ software (https://imagej.net/Fiji, accessed on 29 May 2021). The data represent the means of 10 leaves from each condition. Results were analyzed using a Tukey multiple comparison test (α = 0. 05). Asterisk indicates *p* < 0.05. Error bars indicate SD.

**Figure 9 ijms-22-06804-f009:**
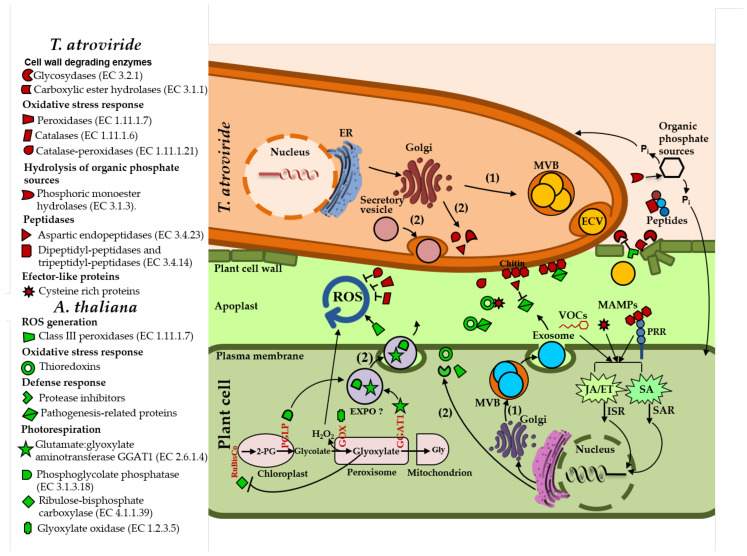
Hypothetical model representing the putative functions of *Arabidopsis* and *T. atroviride* secreted enzymes during their interaction. *T. atroviride* is recognized through its microbe-associated molecular patterns (MAMPs) (i.e., chitin) by pattern recognition receptors (PRRs) localized on the plant cell membrane. *T. atroviride* secretes effector-like proteins and volatile organic compounds (VOCs) into the apoplast. These events activate the JA/ET-dependent ISR-priming and SA-dependent SAR pathways. Subsequently, the accumulation of proteins related to plant defense, including enzymes and proteins with non-enzymatic function, secreted to the apoplast through conventional (outlined with 1) and unconventional secretion pathways that are enhanced (outlined with 2). On the *T. atroviride* side, it secretes an array of enzymes into the apoplast through conventional and unconventional secretion pathways to gain access to the first layers of the root epidermis. These include CWDEs that hydrolyze the plant cell wall polysaccharides. In response, the plant cell secretes protease inhibitors that act against some lytic enzymes liberated by the fungus to contain the cell wall damage. Subsequently, PR proteins with antimicrobial activity are secreted by the plant cell to control the proliferation of the fungus. Afterwards, the plant cell accumulates reactive oxygen species (ROS) into the apoplast, including H_2_O_2_, which is generated by apoplastic class III peroxidases. *T. atroviride* secrete antioxidant enzymes such as catalases and catalase-peroxidases into the apoplast to counteract the oxidative stress caused by ROS. Phosphoric monoester hydrolases are secreted by *T. atroviride* to hydrolyze organic phosphate sources to release inorganic phosphorous (P_i_), which can be taken up by the plant and by the fungus for their metabolism. Furthermore, in response to *T. atroviride*, the plant cell secretes enzymes involved in photorespiration. During photorespiration 2-PG is dephosphorylated in chloroplasts by PGLP to produce glycolate, which is oxidated by GOXs in peroxisomes to produce glyoxylate, which is finally transaminated to glycine (Gly) by the action of GGAT1. PGLP and GGAT1 are secreted by the plant cell, probably by EXPO. The secretion of GGAT1 to the apoplast causes an accumulation of glyoxylate into the cell, which can inhibit the enzyme RuBisCo. Question marks (?) indicate aspects that remain to be clarified. ER, endoplasmic reticulum; ECV, extracellular vesicle.

**Table 1 ijms-22-06804-t001:** Representative predicted enzymes differentially modulated in *T. atroviride* secretome at 24, 48, and 96 h of co-culture with *Arabidopsis*.

JGI Id *	Enzyme Name **	EC No. **	Confidence Score **	Fold Change *** (Log2 Mean)		
Response to oxidative stress				24 h	48 h	96 h
300451	Peroxidase	EC 1.11.1.17	0.72	9.1	1.9	1.1
88379	Catalase-peroxidase	EC 1.11.1.21	1.7	2.1	−1.0	0.5
300992	Thioredoxin-dependent peroxiredoxin	EC 1.11.1.24	1.57	0	3.4	6.9
297668	Catalase	EC 1.11.1.6	2.5	0	−4.9	10.8
94401	Glutathione peroxidase	EC 1.11.1.9	0.76	0	3.3	0.2
215831	Superoxide dismutase	EC 1.15.1.1	0.76	4.7	1.0	0.4
299895	Cytochrome-c peroxidase	EC 1.11.1.5	0.94	0	−2.9	−2.9
155960	Catalase-peroxidase	EC 1.11.1.21	1.7	3.7	−6.5	0.2
298583	Superoxide dismutase	EC 1.15.1.1	2.57	0	−2.5	−2.2
Cell-wall degrading enzymes						
84753	Acetylxylan esterase	EC 3.1.1.72	0.94	6.2	3.8	2.4
297844	Cutinase	EC 3.1.1.74	0.72	5.9	5.9	4.0
296657	Mannan endo-1,6-α-mannosidase	EC 3.2.1.101	1.62	5.6	2.9	4.0
44429	Xyloglucan-specific endo-β-1,4-glucanase	EC 3.2.1.151	0.81	4.9	2.8	4.5
314392	Cellulase	EC 3.2.1.4	0.94	4.4	7.2	10.1
221999	Cellulase	EC 3.2.1.4	0.94	3.1	4.7	5.1
88310	Endo-1,4-β-xylanase	EC 3.2.1.8	0.72	2.4	9.3	3.7
44894	Cellulose 1,4-β-cellobiosidase	EC 3.2.1.91	1.89	4.2	5.4	5.0
88458	Cellulose 1,4-β-cellobiosidase	EC 3.2.1.91	1.89	4.1	5.4	6.9
91075	Glucan 1,3-β-glucosidase	EC 3.2.1.58	0.76	4.8	2.3	0
139054	β-glucosidase	EC 3.2.1.21	0.94	4.7	5.6	10.4
42986	β-glucosidase	EC 3.2.1.21	1.7	−4.8	−5.9	−3.8
223991	β-glucosidase	EC 3.2.1.21	0.94	−6.5	−2.2	−2.9
302027	β-glucosidase	EC 3.2.1.21	0.94	−8.2	−6.6	−0.9
161158	Xylan 1,4-β-xylosidase	EC 3.2.1.37	0.94	−2.3	−2.3	−1.6
161159	Xylan 1,4-β-xylosidase	EC 3.2.1.37	0.94	−4.0	−4.9	−0.9
Proteolysis						
90832	Pepsin A	EC 3.4.23.1	0.76	6.4	3.4	0
142040	Aspergillopepsin I	EC 3.4.23.18	1.81	2.4	3.9	0.2
298116	Aspergillopepsin II	EC 3.4.23.19	0.76	3.9	3.4	0
137451	Penicillopepsin	EC 3.4.23.20	0.81	2.6	6.1	9.2
131866	Penicillopepsin	EC 3.4.23.20	0.86	1.5	3.8	5.2
33651	Rhizopuspepsin	EC 3.4.23.21	0.76	0.6	3.1	6.2
176535	Candidapepsin	EC 3.4.23.24	0.76	3.8	3.5	0
28954	Candidapepsin	EC 3.4.23.24	0.76	2.5	0.1	13.0
292296	Candidapepsin	EC 3.4.23.24	0.76	2.2	0.4	7.5
34007	Candidapepsin	EC 3.4.23.24	0.76	1.6	0.4	9.2
220221	Tripeptidyl-peptidase II	EC 3.4.14.10	0.94	−5.4	−4.4	0
36337	Deuterolysin	EC 3.4.24.39	1.7	0	−1.4	−8.4
40863	Aspergillopepsin I	EC 3.4.23.18	0.81	−2.1	−2.7	0
54917	Dipeptidyl-peptidase II	EC 3.4.14.2	0.72	−3.0	−0.1	0
321810	C5a peptidase	EC 3.4.21.110	0.76	−3.5	0	−10.5
Dephosphorylation						
215617	3-phytase	EC 3.1.3.8	0.76	4.7	0	0.8
44629	Acid phosphatase	EC 3.1.3.2	1.7	2.0	−1.2	0.6
298464	Inositol-phosphate phosphatase	EC 3.1.3.25	1.62	0	4.3	7.1
298832	5′-nucleotidase	EC 3.1.3.5	0.94	−1.4	0.9	4.3
89336	Protein-serine/threonine phosphatase	EC 3.1.3.16	1.7	3.0	0.9	3.9
147790	Phosphoglycolate phosphatase	EC 3.1.3.18	0.81	0	0	4.5

* JGI id numbers were exported from DOE Joint Genome Institute (https://mycocosm.jgi.doe.gov/Triat2/Triat2.home.html, accessed on 19 October 2020). ** According to the analysis of the corresponding protein, the enzyme names, the EC numbers, and confidence scores were extracted from BRENDA database [33]. *** Fold change indicates increased (orange color) or decreased (blue color) *T. atroviride* proteins in co-culture with *Arabidopsis* at 24, 48, and 96 compared with *T. atroviride* growing alone (control).

**Table 2 ijms-22-06804-t002:** Representative predicted enzymes differentially modulated in *Arabidopsis* secretome at 24, 48, and 96 of co-culture with *T. atroviride*.

Locus_Tag *	Gene Symbol *	Enzyme Name **	EC. No. **	Confidence Score **	Fold Change *** (Log2 Mean)		
Response to oxidative stress					24 h	48 h	96 h
At3g49120	PER34	Peroxidase	EC 1.11.1.17	2.70	6.0	6.8	3.1
At2g38380	PER22	Peroxidase	EC 1.11.1.17	2.70	2.2	8.2	7.6
At1g05260	PER3	Peroxidase	EC 1.11.1.17	2.70	2.0	8.2	4.4
At4g11290	PER39	Peroxidase	EC 1.11.1.17	2.70	0	2.6	0
At3g32980	PER32	Peroxidase	EC 1.11.1.17	2.70	0	−1.8	3.8
At3g11630	BAS1A	Thioredoxin-dependent peroxiredoxin	EC 1.11.1.24	3.70	1.9	3.0	0
At2g28190	SODC	Superoxide dismutase	EC 1.15.1.1	3.70	0	2.7	0
At4g25100	FDSD1	Superoxide dismutase	EC 1.15.1.1	2.65	0	4.3	3.7
At5g05340	PER52	Peroxidase	EC 1.11.1.17	2.70	−1.6	−2.8	−2.8
At5g64120	PER71	Peroxidase	EC 1.11.1.17	2.70	−2.3	−4.5	−7.0
At5g64100	PER69	Peroxidase	EC 1.11.1.17	2.70	−3.0	−1.6	−6.7
Defense response							
At4g01610	CATB3	Cathepsin B	EC 3.4.22.1	1.74	0	3.9	0
At3g19390	RD21C	Cathepsin B	EC 3.4.22.1	1.82	0	6.6	3.4
At5g60360	ALP	Cathepsin B	EC 3.4.22.1	0.94	3.6	8.7	6.2
At5g43060	RD21B	Cathepsin B	EC 3.4.22.1	1.82	5.8	0.1	1.6
At1g47128	RD21A	Cathepsin B	EC 3.4.22.1	1.82	0.2	5.6	−2.1
At1g03220	F15K9.17	Chitinase	EC 3.2.1.14	0.81	0	2.5	1.9
At4g16260	At4g16260	Glucan endo-1,3-β-d-glucosidase	EC 3.2.1.39	1.89	4.8	8.4	2.6
At4g19810	CHIC	Chitinase	EC 3.2.1.14	2.70	0	−2.4	−2.1
At2g43620	CHI62	Chitinase	EC 3.2.1.14	2.70	0	−2.8	−1.9
At2g43610	CHI61	Chitinase	EC 3.2.1.14	2.70	−1.4	−3.3	−6.4
At2g43570	CHI	Chitinase	EC 3.2.1.14	2.70	−2.1	−4.9	−9.2
At3g54420	CH5	Chitinase	EC 3.2.1.14	2.70	−2.7	−5.8	−7.8
At1g75040	PR5	Glucan endo-1,3-β-d-glucosidase	EC 3.2.1.39	0.86	−5.5	−6.8	−7.9
Photorespiration							
At5g36790	PGP1B	Phosphoglycolate phosphatase	EC 3.1.3.18	2.70	3.9	6.3	4.4
At1g23310	GGAT1	Glutamate:glyoxylate aminotransferase 1	EC 2.6.1.4	2.70	0	3.5	0
AtCg00490	RBCL	Ribulose-bisphosphate carboxylase	EC 4.1.1.39	3.70	−3.5	−2.4	0

* Locus tags and gene symbols were extracted from The Arabidopsis Information Resources (TAIR) database https://www.arabidopsis.org/ (accessed on 21 September 2020). ** According to the analysis of the corresponding protein, the enzyme names, the EC numbers and confidence scores were extracted from BRENDA database [33]. *** Fold change indicates increased (orange color) or decreased (blue color) *Arabidopsis* proteins in co-culture with *T. atroviride* at 24, 48, and 96 compared with *Arabidopsis* growing alone (control).

## Data Availability

Complete proteomic data were deposited at the PRIDE database, with the accession number PXD023283.

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
