# Peer review of "Secretome Analysis of Arabidopsis–Trichoderma atroviride Interaction Unveils New Roles for the Plant Glutamate:Glyoxylate Aminotransferase GGAT1 in Plant Growth Induced by the Fungus and Resistance against Botrytis cinerea"

_ijms, 2021, doi:10.3390/ijms22136804_

Round 1

Reviewer 1 Report

The authors have done a good work to improve the quality of the paper. In the previous revision I requested for a phenotype of protection, but I can see that it was already in the previous version!! My concerns come from figure 7. With the SD presented in the figure, how the authors can state that there is an induction of protection against B. cinerea? And more applying a Tuckey test, which is quite restrictive test. Perhaps other statistical test? As presented in the figure I do not see any phenotype of protection due to the Trichoderma. At the basal level I agree that GGAT1 may act as a negative regulator of resistance, since the mutant is more resistant. 

Please, clarify this point.

Author Response

Reviewer 1

The authors have done a good work to improve the quality of the paper. In the previous revision I requested for a phenotype of protection, but I can see that it was already in the previous version!! My concerns come from figure 7. With the SD presented in the figure, how the authors can state that there is an induction of protection against B. cinerea? And more applying a Tuckey test, which is quite restrictive test. Perhaps other statistical test? As presented in the figure I do not see any phenotype of protection due to the Trichoderma. At the basal level I agree that GGAT1 may act as a negative regulator of resistance, since the mutant is more resistant. 

Please, clarify this point.

Reply:

We thank the reviewer for the positive comments regarding the new version of our manuscript, and for the invaluable comments/suggestions. In the figure 7C, we are showing the results of challenging and protection phenotype against B. cinerea by showing representative Arabidopsis leaves of two biological replicates with a total of 24 leaves (8 plants) from each condition. The representative leave selected for Col-0 + B. c. condition was changed for a new one where you can appreciate much better such phenotype (see Figure 7C). We agree with the reviewer that the lesion size (cm2) of Arabidopsis leaves (measured by using ImageJ software) of the Col-0 + T. a + B. c. condition show a large SD (Figure 7D) so that was the reason for applying the statistical test. Please take into account that we repeated the experiment twice including 8 plants per condition and from each plant we treated three independent leaves. About the Tukey test, this is a restrictive/stringent test, and the P-value is statistically significant at P<0.05.

Reviewer 2 Report

Review report:

The Manuscript entitled “Secretome analysis of Arabidopsis-Trichoderma atroviride interaction unveils new roles for the plant glutamate:glyoxylate aminotransferase GGAT1 in plant growth induced by the fungus and resistance against Botrytis cinerea” by Gonzalez-Lopez et al., investigated the secretome of Arabidopsis-Trichoderma atroviride co-culture and studied the function of glutamate:glyoxylate aminotransferase  GGAT1 in pathogen response.  

The authors studied the secretome by proteomics LC-MS/MS approach and annotated the differentially accumulated protein classified by plant and fungus. Genetic approach was also employed to study the function of GGAT1 protein. The manuscript is well-written, and the topic is interested to plant community. Considering ggat1-2 mutant does not have strong phenotype, I have a few comments below.

Major comments:

  • For proteomics studies, we normally use three biological replicates but this study only has two replicates.
  • Two independent mutants or genetic complementation should be used for functional validation. The author may include ggat1-1 mutant as well.
  • In Fig. 7, cell death should be measured in addition to monitor lesion size.
  • In Fig.8, H2O2 quantification can be more convencing.

Minor comments:

  • In Line 415, data significance should be defined by p<0.005.
  • In Line 507, gene name should be capitalized,

Author Response

Reviewer 2

Review report:

The Manuscript entitled “Secretome analysis of Arabidopsis-Trichoderma atroviride interaction unveils new roles for the plant glutamate:glyoxylate aminotransferase GGAT1 in plant growth induced by the fungus and resistance against Botrytis cinerea” by Gonzalez-Lopez et al., investigated the secretome of Arabidopsis-Trichoderma atroviride co-culture and studied the function of glutamate:glyoxylate aminotransferase  GGAT1 in pathogen response. 

The authors studied the secretome by proteomics LC-MS/MS approach and annotated the differentially accumulated protein classified by plant and fungus. Genetic approach was also employed to study the function of GGAT1 protein. The manuscript is well-written, and the topic is interested to plant community. Considering ggat1-2 mutant does not have strong phenotype, I have a few comments below.

Major comments:

Comment #1:

For proteomics studies, we normally use three biological replicates but this study only has two replicates.

Reply:

We thank the reviewer for the positive comments regarding our study and for the invaluable comments/suggestions. In our work we included several time points, which represent independently sampled cultures (here, hydroponic culture containers containing 30 plants each), as we have done (see lines 168-182 for details of the filtering criteria). This approach allowed the identification of a higher number of proteins of both organisms compared with other studies that have analyzed the interaction between plants and symbiotic fungi.  Furthermore, the presented results were considered as positives if such proteins were identified in the two replicates (see lines 174-176).

Comment #2:

Two independent mutants or genetic complementation should be used for functional validation. The author may include ggat1-1 mutant as well.

Reply:

The phenotype for the ggat1-1 mutant line was described in detail previously by Verslues, et al. (2007, PMID: 17318317). Under stress conditions (low water potential and NaCl stresses), this mutant showed decreased levels of abscisic acid (ABA). ABA deficiency induced the up-regulation of JA-ET responsive genes and correlates with enhanced resistance in Arabidopsis to necrotrophic pathogens (see PMID: 15548743) (discussed in lines 805-813). Based on these data, it is likely that ggat1-1 will show enhanced resistance to B. cinerea like ggat1-2. Other studies support this hypothesis. Lian and coworkers (2018) generated a set of ggta1 mutants (CTC-deletion and T-deletion) using the Ecotype Arabidopsis thaliana Landsberg erecta (Ler-0) by using CRISPR/Cas9 technology, and no mutations were detected at the possible off target site. Basically, they observed that both mutants have similar photorespiration phenotypes (PMID: 29392632) as reported for ggat1-1 mutants in Arabidopsis thaliana Col-0 Ecotype (Verslues, et al. 2007, PMID: 17318317). Similar photorespiratory phenotypes were reported as well for T-DNA insertional lines GK-649H07 and GK-847E07 which bear the T-DNA at the second exon and at the eight exon, respectively (Dellero et al., 2015; PMID: 26216646). Taken together all these results reported in the literature, we think that potentially the different alleles will show similar phenotypes when challenged against B. cinerea as well as in those plants previously treated with T. atroviride. In addition, ggat1-1 was already characterized for hydrogen peroxide production under control condition (Verslues, et al. 2007, PMID: 17318317). Therefore, we do consider these data to compare and explain our results. It might be helpful to evaluate the response of ggat1-1 to both Trichoderma and B. cinerea, however these mutant alleles are not available in TAIR, which will complicate your request. In addition, asking for the mutants to the authors, their genotyping and perform the experiments could take at least 8 months. This only in case the authors accept to share their mutant alleles with us. Furthermore. the restrictions to import biological materials from abroad have been hardened in Mexico since the pandemic provoked by SARS-CoV-2 started.

Comment #3:

In Fig. 7, cell death should be measured in addition to monitor lesion size.

Reply:

We agree with the reviewer that the measuring of cell death would help to improve our manuscript, however, our main goal was to test the role of one of the differential accumulated proteins in response to T. atroviride and against its challenging against B. cinerea. As you can see at figure 7C and D, our results show clearly what we wanted to demonstrate. Furthermore, most of the published works showing plant challenging against B. cinerea present their data as we show in this work (i. e., see PMID: 29076546, PMID: 29866036, PMID: 25677379, PMID: 22128140 and PMID: 32990799). That is, only the phenotype of leaves in response to B. cinerea, and the quantification of the damaged area. For this reason, we do not consider necessary to perform additional experiments. Nevertheless, we decided to work on the major criticisms that were raised by the reviewers for a previous revised version of this manuscript. We have done considerable experimental work in response to the reviews of the first version (which was now resubmitted formally as new manuscript ID number ijms-1240257 due to the time needed). This is reflected in the overall favorable re-evaluation by Reviewer 1.

Comment #4:

In Fig.8, H2O2 quantification can be more convencing.

Reply:

Most of the published works showing DAB staining results present their data with images of plant leaves and ImageJ quantification from DAB-stained leaves (i. e. see PMID: 32706394 and PMID: 29352810). In this new version of the manuscript, we present DAB intensity quantification/area (mm2) of Arabidopsis leaves using Fiji-ImageJ software and t-test analysis (see Figure 8B). We hope you find now suitable our results as analyzed.

Minor comments:

Reviewer #2

In Line 415, data significance should be defined by p<0.005.

Reply:

We corrected the data significance (p<0.05) (see line 415) because the analysis of plant growth stimulation by Trichoderma was done using a Tukey multiple comparison test defined by p<0.05.

Reviewer #2

In Line 507, gene name should be capitalized, THPG1.

Reply:

The gene name thpg1 now is in line 519. In filamentous fungi like Neurospora crassa and Trichoderma spp., for the genes, and proteins nomenclature we use to write the first in italic lower-case letters, whereas the proteins are written in nonitalic capital letters. Please see NEUROSPORA GENETIC NOMENCLATURE. Fungal Genetics Newsletter 46:31-41 for nomenclature rules.

Round 2

Reviewer 2 Report

The authors have provided plausible responses to my comments. I have no further requests.